# A New Assessment of Robust Capuchin Monkey (*Sapajus*) Evolutionary History Using Genome-Wide SNP Marker Data and a Bayesian Approach to Species Delimitation

**DOI:** 10.3390/genes14050970

**Published:** 2023-04-25

**Authors:** Amely Branquinho Martins, Mônica Mafra Valença-Montenegro, Marcela Guimarães Moreira Lima, Jessica W. Lynch, Walfrido Kühl Svoboda, José de Sousa e Silva-Júnior, Fábio Röhe, Jean Philippe Boubli, Anthony Di Fiore

**Affiliations:** 1Centro Nacional de Pesquisa e Conservação de Primatas Brasileiros, Instituto Chico Mendes de Conservação da Biodiversidade, Cabedelo 58310-000, PB, Brazil; 2Primate Molecular Ecology and Evolution Laboratory, Department of Anthropology, The University of Texas at Austin, Austin, TX 78712, USA; 3Laboratório de Biogeografia da Conservação e Macroecologia, Instituto de Ciências Biológicas, Universidade Federal do Pará, Belém 66077-530, PA, Brazil; 4Institute for Society and Genetics, Department of Anthropology, University of California-Los Angeles, Los Angeles, CA 90095, USA; 5Instituto Latino-Americano de Ciências da Vida e da Natureza, Centro Interdisciplinar de Ciências da Vida, Universidade Federal da Integração Latino-Americana, Foz do Iguaçu 85870-650, PR, Brazil; 6Museu Paraense Emílio Goeldi, Ministério da Ciência, Tecnologia, Inovações e Comunicações, Coordenação de Zoologia, Campus de Pesquisa, Setor de Mastozoologia, Belém 66077-830, PA, Brazil; 7Laboratório de Evolução e Genética Animal, Universidade Federal do Amazonas, Manaus 69067-005, AM, Brazil; 8School of Science, Engineering and the Environment, University of Salford, Salford M5 4WT, UK; 9Tiputini Biodiversity Station, Universidad San Francisco de Quito, Quito 170901, Ecuador

**Keywords:** Neotropical primates, phylogenomic, ddRADseq, evolutionary history

## Abstract

Robust capuchin monkeys, *Sapajus* genus, are among the most phenotypically diverse and widespread groups of primates in South America, with one of the most confusing and often shifting taxonomies. We used a ddRADseq approach to generate genome-wide SNP markers for 171 individuals from all putative extant species of *Sapajus* to access their evolutionary history. Using maximum likelihood, multispecies coalescent phylogenetic inference, and a Bayes Factor method to test for alternative hypotheses of species delimitation, we inferred the phylogenetic history of the *Sapajus* radiation, evaluating the number of discrete species supported. Our results support the recognition of three species from the Atlantic Forest south of the São Francisco River, with these species being the first splits in the robust capuchin radiation. Our results were congruent in recovering the Pantanal and Amazonian *Sapajus* as structured into three monophyletic clades, though new morphological assessments are necessary, as the Amazonian clades do not agree with previous morphology-based taxonomic distributions. Phylogenetic reconstructions for *Sapajus* occurring in the Cerrado, Caatinga, and northeastern Atlantic Forest were less congruent with morphology-based phylogenetic reconstructions, as the bearded capuchin was recovered as a paraphyletic clade, with samples from the Caatinga biome being either a monophyletic clade or nested with the blond capuchin monkey.

## 1. Introduction

Capuchin monkeys (genus *Cebus sensu lato*) (Erxleben, 1777) are a lineage of primates that are considered one of the groups with the most confusing taxonomy among Neotropical mammals [1,2]. The group has been divided into two genera—*Cebus* (the gracile capuchins) and *Sapajus* (the robust capuchins)—based on several traits, including genetic data [3,4,5]. Robust capuchin monkeys (*Sapajus*) have a widespread distribution found throughout South America, from the Colombian Llanos, through the Amazon basin, to the Cerrado, Caatinga, Atlantic Forest, and Pantanal of Brazil, and in countries of the Southern Cone [6,7].

Over the past quarter century, taxonomists and phylogeneticists have suggested a host of arrangements for robust capuchin taxonomy (Appendix A) [1,3,5,7,8,9,10,11,12,13], with different studies dividing the genus into either four [5,11], six [13], seven [3], or eight [7,14] species. Under the broad “eight-species” classification, the taxonomic arrangement includes the following: *Sapajus xanthosternos*, the yellow-breasted capuchin, which is endemic to Brazil and found in the Atlantic Forest and Caatinga, from the south and east of the São Francisco River to the north of the Jequitinhonha River; *Sapajus robustus*, the robust tufted capuchin, which is endemic to the Brazilian Atlantic Forest, occurring at the south of the Jequitinhonha River and extending as far as the Doce River at the south and the Serra do Espinhaço mountains to the southwest; *Sapajus nigritus*, the black-horned capuchin, which occurs from the Doce River in Minas Gerais, Brazil, extending through the southern region of Brazil and the extreme northeastern tip portion of Argentina provinces of Iguazú and Misiones, at the east of the Paraná River; *Sapajus apella*, the brown capuchin, and *Sapajus macrocephalus*, the large-headed capuchin, both of which are Amazonian species; *Sapajus cay*, Azara’s capuchin, which ranges through the northern tip of Argentina, southern Bolivia, the eastern half of Paraguay, and Brazil, extending close to the Amazon in the forests of the Pantanal and the intersection of this biome and the Cerrado; *Sapajus libidinosus*, the bearded capuchin, which occurs throughout the Cerrado and Caatinga biomes of Brazil and has a wide distribution in these dry, savanna-like habitats; and *Sapajus flavius*, the blond capuchin, which has the smallest geographic range and is mostly limited to sparse remnants of the Atlantic Forest in the northeast of Brazil [7] (Figure 1). To the north and east, the Atlantic Ocean limits the distribution of this species. The western limit of its distribution is undefined, but supposedly coincides with areas of transition between the Atlantic Forest and the drier Caatinga biome [15,16,17], where it potentially overlaps with the distribution of the bearded capuchin (*S. libidinosus*). A lack of information about populations of wild capuchins in the Atlantic Forest–Caatinga transition has precluded researchers from clearly defining the geographic limits between *S. libidinosus* and *S. flavius*, generating uncertainties about their taxonomic identity.

As the blond capuchin monkey occurs mostly in the coastal Atlantic Forest, which is the biome where the robust capuchins are believed to have first speciated, it would be rather easy to hypothesize the blond capuchins should be more closely related, phylogenetically, to other capuchin species from the same biome, such as *S. xanthosternos*, the yellow-breasted capuchin. However, recent studies have suggested that blond capuchin monkeys either belong to the same species as, or are a sister taxon to, the bearded capuchin monkey, *S. libidinosus* from the Cerrado, and these two species are both more closely related to robust capuchin monkeys from the Amazon than to other species occurring in the Atlantic Forest [5,13]. Thus, several studies have recovered *S. flavius* as either a monophyletic clade within the widespread Amazonian group of robust capuchins (along with *S. apella*, *S. macrocephalus*, *S. cay*, and *S. libidinosus*) or as a sister taxon to *S. libidinosus*, the bearded capuchin, to the exclusion of the Amazonian clade. These studies have used sequence data from mitochondrial and certain nuclear markers (ultraconserved elements, UCEs) to infer the robust capuchin monkey phylogeny, but they only included three samples of the blond capuchin monkeys, all of which came from populations inhabiting the extreme eastern portion of the species’ distribution in the Atlantic Forest [5,13]. Additionally, limitations with the utility of both these genetic markers have been reported, especially for recently evolving lineages [18,19], and increasing taxon sampling, especially from populations living closer to the species boundaries, is sorely needed to provide a more complete picture of robust capuchin phylogenetic history [20,21,22]. Thus, one of the key goals of this study is to reconstruct the phylogenetic relationships and phylogeographic history of the capuchin monkeys of the northern Atlantic Forest and Caatinga biomes of Brazil and describe how they fit within the *Sapajus* phylogeny.

Understanding evolutionary history and the underlying processes that drive rapid radiations is an important goal in evolutionary biology [23,24,25,26]. Therefore, given the uncertainties in the *Sapajus* phylogeny (especially within the recently evolving widespread lineages in the Amazon, Cerrado, Caatinga, and extreme northeastern Atlantic Forest of Brazil), better characterization of the evolutionary history of the *Sapajus* radiation is of great interest. However, establishing phylogenetic relationships among lineages that have undergone rapid and recent diversification is a challenge due to incomplete lineage sorting, which may be due to genes that have evolved slowly relative to the rate of speciation [27,28,29,30]. Additionally, it is especially difficult to reconstruct evolutionary history for geographically widespread lineages where there are few barriers to dispersal, where multiple zones of contact between lineages may exist, and where reproductive isolation is incomplete [31,32,33,34].

Fortunately, utilizing information from hundreds or thousands of genomic markers, such as single nucleotide polymorphisms (SNPs), modern phylogenomic analysis has the potential to decrease the impact of such difficulties. Considering the increasing availability of high-throughput sequencing technologies and their rapidly decreasing costs, it is now feasible to study the evolutionary history of lineages at the genome-wide scale for many taxa of interest, including non-model organisms [35,36,37]. Several methods have been developed to discover and screen a set of genome-wide markers by subsampling and sequencing just a small fraction of the genome, which, nonetheless, can include tens of thousands of variable sites [38,39,40,41]. Restriction-Site-Associated DNA sequencing (RADseq) is a term applied to a group of next-generation sequencing methods that rely on the use of one or more restriction enzymes to cleave the genome into a set of short DNA fragments flanked by the restriction sites [39], just a fraction of which are then isolated (e.g., by size) and sequenced. This process allows for genome-wide marker discovery and typing at a high coverage and low cost, favoring markers to be genotyped accurately across individuals at different population or taxonomic levels [42,43,44]. RADseq-based methods have been successfully used to discover thousands of SNPs in phylogenetically diverse organisms including fish [45,46,47], insects [48,49], birds [50], and mammals [51,52], including primates [43,53,54,55].

For this study, we constructed reduced representation genomic libraries using a variant of the RADseq approach known as the “double-digest Restriction-Site-Associated DNA” (ddRAD) method, coupled with Illumina high-throughput sequencing, which allowed us to genotype individual samples by sequencing short fragments of DNA flanked by a specific combination of two restriction endonuclease recognition sites [56]. This approach allowed us to genotype thousands of informative single nucleotide polymorphisms (SNPs) sampled from across the capuchin genome to infer the phylogenetic relationships among taxa in the *Sapajus* radiation and to assess the congruence of our results with previous phylogenetic studies of the genus. Our study used a wide range of samples from individuals representing all putative species of *Sapajus*, most with known provenance. In addition to a set of individuals also used in previous studies (e.g., [13]), we acquired dozens of new samples from the yellow-breasted, the bearded, and the blond capuchin monkeys, including samples from deep within the geographic ranges of each of these two later species, as well as from populations of capuchins occupying areas of the Atlantic Forest–Caatinga transition, whose taxonomic assignment is uncertain. Therefore, this study includes both a larger number of samples and samples from more localities than any previous research on *Sapajus* phylogenetics. To infer the phylogenetic relationships among our *Sapajus* samples, we used both a Maximum Likelihood (ML) and a multispecies coalescent approach grounded in quartet-based phylogenetic inference that combines information from multiple gene trees. In addition, we used a Bayes Factor (BF) validation method to test among seven alternative hypotheses of species delimitation within the *Sapajus* radiation. Finally, due to our improved sampling of the blond capuchin monkey, we also give special attention to this species and its phylogenetic relationships with its closest congener by testing alternative hypotheses regarding the species delimitation of both blond and bearded capuchin monkey lineages and by evaluating whether the presence of the blond capuchin monkey in the Caatinga biome is corroborated by our phylogenomic analyses.

## 2. Materials and Methods

### 2.1. Samples

For this study, we used samples from a total of 171 individuals (Figure 2): 149 individuals from the 8 putative species of the genus *Sapajus*, 19 individuals from 3 species of the genus *Cebus*, and 3 individuals from the genus *Saimiri* (Appendix A). Provenance is known for 159 of these samples. Overall, 109 of these samples were collected by collaborators and used in previous studies [5,13,57], while 62 were newly collected for this study. For the latter set of samples, we collected blood or tissue samples from seven populations of wild blond capuchins, six populations of bearded capuchins, and one population of yellow-breasted capuchins in a series of field seasons between 2005 and 2016. Animals were captured in tomahawk traps baited with corn and sedated with ketamine HCl (~30 mg/kg, IM) in consultation with a veterinarian. Traps were kept open during the day and monitored regularly (~every 2 h), and captured animals were processed immediately to minimize holding time. Individuals were released as soon as the effects of the sedation wore off. During processing, a wildlife veterinarian collected 3 to 5 mL of blood (depending on body mass) from the femoral vein of each individual using a 23-gauge needle. Samples were collected into color-coded vacutainers containing EDTA as an anticoagulant and then preserved on ice. At the end of each day’s trapping, blood and tissue samples were transferred to −20 °C for long-term storage. Additionally, health conditions were examined, and photographs, morphometric measurements, weight, and dental casts were collected while animals were sedated.

During each capture event, all team members involved in handling animals used gloves and face masks as precautions against disease transmission from researchers to animals and zoonotic infection. Animal immobilization procedures were conducted by Brazilian wildlife veterinarians with broad experience in primate fieldwork. Capture procedures, measurements, and sample collection used were approved by the Brazilian SISBIO/ICMBio (Research Permit Number 19927), and all protocols used were approved by the Institutional Care and Use Committee (IACUC) of the University of Texas at Austin (Protocol ID: AUP-2016-00077). Additionally, DNA, blood, or tissue samples for other *Sapajus*, *Cebus* (outgroup), and *Saimiri* (outgroup) species were obtained from collaborators (Appendix A).

### 2.2. DNA Extraction and Quantification

For most samples newly collected in this study, DNA extraction was performed in the Primate Molecular Ecology and Evolution Laboratory at the University of Texas at Austin, although, for some individuals, genomic DNA was provided by colleagues. Fresh genomic DNA was extracted from the tissue and blood samples using the DNeasy Blood & Tissue Kit^®^ (Qiagen, Germantown, MD, USA) as per the manufacturer’s instructions. The nucleic acid concentration of all samples was quantified in the Institute for Cellular and Molecular Biology’s DNA sequencing facility at the University of Texas at Austin using a Qubit 2.0 fluorometer (LifeTechnologies, Carlsbad, CA, USA). Most of the samples yielded sufficient genomic DNA for normalization (via dilution) to ~10 ng/μL for digestion and subsequent library construction.

### 2.3. Whole Genome Markers

We constructed reduced representation genomic libraries using the double-digestion Restriction-Site-Associated DNA method (ddRADseq) [56]. After normalization, a total of 100 ng of genomic DNA for each sample was first digested with two restriction enzymes, *SphI* and *MluCl*, that have been previously demonstrated as useful for discovering large numbers of variable SNP markers across platyrrhines [58]. The resultant DNA fragment libraries were then purified using AMPure Bead XP and had P1 and P2 Illumina adapters and barcodes ligated to the fragments. Samples were pooled, size selected using Sage Science Pippin-Prep to focus only on fragments of 300 ± 30 bp, and re-amplified using Phusion High Fidelity PCR. Library quality was assessed using an Agilent 2100 Bioanalyzer (Agilent, Santa Clara, CA, USA), and samples were sequenced using the Illumina HiSeq 4000 (Illumina, Inc., San Diego, CA, USA) platform with paired-end reads of 2 × 150 bp length and a minimum of 2–3 million reads for each sample. All library preparation and sequencing were performed at the University of Texas at Austin’s core Genome Sequencing and Analysis Facility.

### 2.4. Quality Control

Raw sequencing reads were first quality checked using FASTQC 0.11.9 [59], which is a quality control application specifically for high-throughput sequence data. Reads were then filtered using the BBDuk software from the BBTools suite of bioinformatic tools, version 38.76 [60]. With this tool, reads were first adapter-trimmed at the 3′ end using a *kmer* length of 22, allowing a maximum of 3 mismatches, and discarding any reads smaller than 30 bp. Trimming was performed with the “tbo” and “tpe” options in order to trim adapters based on pair overlap detection and to trim all reads to the same length when an adapter sequence was only detected in one read of a pair. Then, as PhiX DNA is commonly used as a spike-in control during library preparation for Illumina sequencing, all reads that mapped to the PhiX genome were filtered out. After verification of the correct pairing of R1 and R2 reads, all unpaired sequences were discarded from further analysis. Because read quality often decreased at the end of a read, reads were then trimmed from the terminal end back to the first base that had an average FASTQC quality score of Q < 30. Lastly, we discarded all reads with an average quality score of Q < 30.

All remaining raw reads were assigned to individual samples using their barcode through the deML software [61], allowing for up to one mismatch in the barcode sequence. The final set of filtered, trimmed, and assigned reads consisted of, at most, 145 bp reads, beginning with either the 4 bp *MluCl* (for the R1 read) or the 6 bp *SphI* (for the R2 read) restriction enzyme recognition sites. We then ran a second round of adapter trimming at the 5′ end of each read to remove the additional 5 or 4 bases corresponding to the restriction enzyme recognition sites for the R1 and R2 reads, respectively. Finally, all bases with Phred quality scores of less than 20 were replaced with the uncalled base symbol (i.e., Ns), and reads with more than 5% Ns were discarded from further analysis. Overall, ~95% of the raw reads were kept after all trimming and filtering steps.

### 2.5. Assembly and SNP Calling

Identification of orthologous ddRAD sequences and SNP loci was performed using the software ipyrad [v.0.9.19] [62], which is a free open source tool for assembling restriction site-associated DNA sequence datasets. Using ipyrad, paired-end reads were mapped to the *Sapajus apella* reference genome [63] (NCBI, BioProjects accession PRJNA717806), identifying read copies from the same locus within samples and producing gapped alignments. Paired reads that mapped with incorrect orientation or to multiple locations (paralogous sequences) were discarded. The final set of putative loci for each sample was generated from those clusters of reads—i.e., groups of highly similar sequences mapping to the same genome location—with a sequencing depth of at least six reads (≥6x) [64]. We set the maximum number of heterozygous sites (Hs) and Ns within the consensus sequence at each locus as the upper bound of the 95% CI of these two variables estimated from the array of consensus sequences. To cluster both within and across samples, we set the clustering threshold to 85% according to previous studies (e.g., [58,65]) that have demonstrated that over-splitting tends to occur when using more stringent clustering thresholds. Such over-splitting in the identification of putative loci can be detrimental for phylogenetic inference [66]. Consensus loci found within each sample were then aligned across samples using Muscle v3.8.31 (EMBL-EBI, Hinxton, Cambridgeshire, UK) [67] to generate an initial data matrix of all putative homologous loci that were recovered in at least four individuals. We applied two additional filters to generate the final dataset, while avoiding ambiguous genotypes for each sample. First, we discarded putative loci that were heterozygous in more than 50% of individual samples to avoid potentially clustering paralogous loci rather than true heterozygous sites. Second, we excluded loci containing more than a specified maximum number of SNP sites across the entire set of samples to avoid potential effects of poor alignments in repetitive regions. The threshold for this maximum number of SNPs was set as the upper bound of the 95% CI of the distribution of the number of SNPs per locus. Bioinformatic analyses were undertaken at UT’s Texas Advanced Computer Center (TACC).

Finally, to evaluate the effects of missing data due to allelic dropout, low sequence coverage, or the random effect of next generation sequencing, we created three different datasets with varying levels of locus coverage across samples. That is, we created four progressively smaller data matrices comprising all the loci that were present in at least ≈ 30%, 60%, 75%, and 90% of individuals.

### 2.6. Phylogenetic Analysis

We used two different methods to infer the phylogenetic relationships among the set of *Sapajus* and outgroup samples. First, we conducted a Maximum Likelihood (ML) analysis using the concatenated RAD sequence data from all loci in the final genotype matrix [68]. Second, we used a coalescent-based approach using quartet-based phylogenetic inference under a Multispecies Coalescent (MSC) theory framework [69,70,71,72].

Model selection was performed for the whole concatenated dataset using ModelFinder [73] implemented in the IQ-TREE v. 1.6.12 software [74,75], using the corrected Akaike’s information criterion (AICc). All nucleotide substitution models supported by IQ-TREE were tested. We then used the best fitted substitution model for phylogenetic tree reconstruction under the ML framework in the IQ-TREE v. 1.6.12 software [74,75]. For the ML analysis, 500 initial independent searches were done on the original alignment. Each search started from 100 parsimony trees; then, the 20 best scoring parsimony trees were selected to optimize the search, and only the 5 top scoring trees were retained in the candidate set to improve ML tree search efficiency. Searches were run using default tree search parameters in IQ-TREE. Node support was assessed with 1000 standard nonparametric bootstrap replicates [76]. For the MSC analysis, we used the SVDquartets approach [72] to estimate the species tree. SVDquartets, which is a single-site quartet framework, exhaustively samples subsets of four individuals from the data matrix to produce the best quartet tree and then constructs a species tree from all sampled quartets. We implemented the SVDquartets species tree inference using the program Tetrad v0.9.13, within the ipyrad analysis toolkit [62]. Statistics of support for each node were estimated in Tetrad through 1000 bootstrap replicates by resampling the number of loci with replacement to the same size as the original dataset. Phylogenetic analyses were undertaken using the Lonestar high-performance mainframe computing infrastructure at UT’s TACC.

### 2.7. Bayesian Analysis and Divergence Dating

In order to more robustly infer species boundaries and evaluate the validity of the various monophyletic groups recovered in our ML and MSC analyses, we used a validation method to test alternative hypotheses of species delimitation based on the assignment of samples to candidate species, which is a robust approach, as it explicitly models the process of lineage diversification among a set of presumed candidate lineages [77]. Statistically delimiting species boundaries using multi-locus or genome-scale DNA sequences is increasing an objective of certain taxonomic analyses—especially for identification of cryptic species—and is a prominent research field [78,79,80]. In addition, model-based genome-wide approaches using the MSC framework are advantageous, as they account for coalescent processes when estimating phylogenetic relationships while, at the same time, considering variation in demographic parameters, variation in molecular sequences, and incomplete lineage sorting [79,81]. 

Overall, we tested eight alternative species delimitation models (H0 to H7) (Figure 3). Model H0 was our null hypothesis, which considers all *Sapajus* samples as belonging to a single species. All the other models (H1 to H7) considered, as species, *S. xanthosternos*, the yellow-breasted capuchin, *S. robustus*, the crested capuchin, and *S. nigritus*, the black-horned capuchin, with varying assumptions regarding other putative forms of *Sapajus*, taking into account the fact that the most recent phylogenetic reconstructions of the *Sapajus* radiation [5,13], as well as our own ML and MSC analyses, consistently divide the Atlantic Forest capuchin forms into three different monophyletic clades. Model H1 posits a widespread *S. apella* species, which includes all but those three forms mentioned above, in a single species, as suggested by recent mtDNA- and UCE-based analyses [5,13]. Model H2 posits *S. flavius* and *S. libidinosus*—with all samples from Cerrado and Caatinga—as separate species, plus a widespread Amazonian clade that includes *S. cay*, *S. macrocephalus*, and *S. apella*, which was suggested by SNPs recovered from a UCEs analysis [13]. Model H3 posits a situation similar to H2 but considers *S. flavius* and *S. libidinosus* as the same species. Models H4 and H5 were derived from the results of our IQ-TREE analysis; H4 is similar to H3 but considers *S. cay* as a separate species from the other Amazonian forms (*S. macrocephalus* and *S. apella*), while H5 considers as separate species all monophyletic clades found in our IQ-TREE analysis. This hypothesis places *S. flavius* with samples from the Caatinga and *S. libidinosus* (*S. libidinosus* 1 = samples from Serra da Capivara + samples from Maranhão; *S. libidinosus* 2 = samples from the state of Goiás—southern Cerrado) as one species. Finally, models H6 and H7 were derived from the results of our MSC analysis. These hypotheses, similar to models H4 and H5, also posit *S. flavius* (samples from the Atlantic Forest and a couple of localities from Caatinga) and two *S. libidinosus* species (*S. libidinosus* 1 = samples from Caatinga; *S. libidinosus* 2 = samples from Cerrado) as one species. H6 and H7 differ, however, in their consideration of Amazonian *Sapajus*, with H6 positing that all the Amazonian forms belong to one unique widespread taxon, while H7 considers *S. cay*, *S. apella* 1 (the northeastern Amazonian clade from ML and MSC phylogenetic reconstructions), and *S. apella* 2 (the southwestern Amazonian clade from ML and MSC phylogenetic reconstructions), as three separate taxa.

The Bayes Factor (BF) has been widely used as a model selection tool when comparing alternative models or phylogenetic hypotheses [78,80,82,83]. A Bayes Factor is calculated as the ratio of the marginal likelihood of one model to the marginal likelihood of a competing model, where the marginal likelihood measures the average fit of a model to the data [84,85]. For this study, we used a method that simultaneously estimates the species tree while evaluating alternative species delimitation models by implementing the Bayes Factor Delimitation of Species (BFD) algorithm [80] in StarBeast2 v0.15.5 [86] within the software BEAST2 [87]. The marginal likelihood was estimated for each model through Path Sampling (PS) using BEAST2’s [87] PathSampleAnalyser package application. Path sampling has been shown to generate highly accurate results for model selection of species delimitation [80,88]. PS was run for a chain length of 25 million generations for 20 path steps (totaling 500 million generations). All BFD StarBeast2 analyses were performed assuming a strict clock model, using the HKY site model with four gamma categories, and a Birth–Death speciation model. The convergence of the runs was assessed using ESS parameters with Tracer v1.7 [89]. We used marginal likelihood values to rank model hypotheses H1 to H7 and Bayes Factors to estimate the support for each model relative to the model with the highest marginal likelihood. The strength of support from Bayes Factor (BF) estimates for competing model hypotheses was evaluated according to Kass and Raftery’s (1995) [84] framework. Therefore, the BF scale was used as follows: (a) 0 < BF < 2 means “not worth more than a bare mention”, 2 < BF < 6 means positive evidence, 6 < BF < 10 means strong support, and BF > 10 means decisive support to distinguish between competing species delimitation model hypotheses. BFD analyses were run twice to confirm the consistency between runs. A final tree for the higher ranked model hypothesis was then obtained by combining posterior replicates with LogCombiner [87] and summarizing with TreeAnnotator v2.6 [87] under the maximum clade credibility trees and excluding 20% burn-in. Bayesian analyses were done using high-performance computing facilities at CIPRES [90] and TACC.

Finally, evolutionary timescale and confidence intervals for divergence dates for the capuchin radiation were inferred using a Bayesian MCMC method using the software StarBeast2 v0.15.5 [86] implemented in BEAST2 [87], assuming a strict molecular clock, using the constant population size model, using an HKY site model with four gamma categories, under the Birth–Death model prior for lineage branching, and using default hyperpriors. We used one calibration point on the root node to obtain the posterior distribution of the estimated divergence times: the divergence of *Saimiri* and capuchin monkeys (*Cebus* and *Sapajus*) based on the two fossil records of *Neosaimiri fieldsi* and *Panamacebus* [91,92]. Therefore, considering the minimum and maximum date estimates for these fossil records, we ran three divergence time analyses according to the calibration models proposed by [93], as follows: (a) model calibration 1: an exponential distribution for the prior with a predefined offset of 12.1 Ma; (b) model calibration 2: a lognormal distribution with an offset of 12.5 Ma, mean of 1.8 million years, and standard deviation of 0.4 million years; and (c) model calibration 3: a lognormal distribution with an offset of 20.0 Ma, mean of 2.0 million years, and standard deviation of 0.5 million years.

## 3. Results

### 3.1. Molecular Data

Overall, we generated ~2 × 109 raw reads for all the samples spread across 18 pooled libraries each containing between 19 and 24 samples. All libraries showed good quality, with mean Phred scores of 39 or above for R1 and R2 reads. As expected for the next generation sequencing approach, individual base quality scores decreased at the end of the reads, with the last 5 bp having mean Phred scores of 36 and 30 for the R1 and R2 reads, respectively. Approximately 95% of the raw reads were kept after all BBduk adapter trimming, PhiX removal, and quality filtering and trimming steps (Appendix A), and approximately 99% of these reads were successfully assigned to an individual sample in the demultiplexing step.

An average of 58,340 loci with depth >6x (min. = 9870, max. = 201,459) were assembled within each individual sample after all quality filtering and trimming steps (Appendix A). The number of total loci recovered in the final genotyping matrix (across all samples) varied according to the criterion chosen for the number of samples a locus had to be present in for its inclusion in the final genotype matrix (Table 1). Overall, the number of total loci in the final ipyrad genotype matrices varied from 16,880 to 64,081 for the matrices with ≈20% (minimum of 150 samples used to select a locus) to ≈40% (minimum of 50 samples used to select a locus) missing data, respectively (Table 1).

### 3.2. Sapajus Phylogenetic Reconstructions

Both ML and MSC analyses recovered the same tree topology when varying the degree of missing data. Results shown for ML and MSC are for the analysis with up to 34% missing data (i.e., genotype data available for a minimum of 50 samples for a locus to be included in the final genotyping matrix).

As expected from previous studies, we recovered strong support (100% of bootstrap replicates) for reciprocal monophyly of the *Sapajus* and *Cebus* genera from both the ML and MSC analyses (Figure 4, Appendix A). Within *Sapajus*, our ML analysis recovered *S. robustus* and *S. xanthosternos* as sister clades, with these two groups closer to *S. nigritus*, to the exclusion of all other clades, with full bootstrap support. Our ML phylogenetic reconstruction also recovered a monophyletic clade comprising *S. flavius* and *S. libidinosus*, to the exclusion of the capuchin monkeys occurring in the Pantanal and Amazon biomes, which were recovered as three monophyletic taxa, including *Sapajus cay* and two other Amazonian clades, all with strong support (100% bootstrap), see Figure 4. However, one of the samples from *S. cay* was recovered within one of the Amazonian clusters (Appendix A). Complementarily, our MSC analysis also recovered *S. robustus*, *S. xanthosternos*, and *S. nigritus* as monophyletic groups, with *S. robustus* as the sister group to all *Sapajus* with strong support (100%), while the support values for the split of the monophyletic clades comprising either *S. xanthosternos* or *S. nigritus* were more moderate (80% and 89%, respectively). The MSC phylogenetic reconstruction (Figure 4) also recovered a monophyletic clade comprising *S. flavius* and *S. libidinosus* (100% support) and a grouping of three additional monophyletic clades, *S. cay* (with moderate support of 86%) and two Amazonian forms with strong support (100%). Again, as in the ML reconstruction, one of the samples putatively assigned to *S. cay* was recovered within one of the Amazonian clusters. Therefore, both ML and MSC analyses recovered *S. robustus* and *S. xanthosternos* as early splits within the *Sapajus* radiation and *S. nigritus* as the sister group to all other *Sapajus* clades. Both analyses also recovered Amazonian *Sapajus* as divided into a northeastern clade and a southeastern group, with maximum support in both analyses.

Overall, the species trees recovered from both the ML and MSC analyses were congruent in recovering mostly the same set of monophyletic clades. However, aside from the support values, the main difference between our ML and MSC analyses was regarding the placement of the samples identified as *S. flavius* and *S. libidinosus* species. While both analyses recovered strong structure within the clade comprising samples from both *S. flavius* and *S. libidinosus* (with the latter being paraphyletic), the assignment of individual samples to the reciprocally monophyletic clades recovered was not congruent between analyses. The ML phylogenetic analysis suggests that samples of *S. libidinosus* from the southern Cerrado (Goiás state) belong to a different clade than samples of *S. libidinosus* from the northern Cerrado (Maranhão and Piauí states), while it placed all *S. libidinosus* from the Caatinga biome as closer to *S. flavius*, within a structured monophyletic group of *S. flavius* + *S. libidinosus* from the Caatinga (Figure 4, Appendix A). That is, the ML-based inference of phylogenetic relationships within the *S. flavius* + *S. libidinosus* group does not agree with that expected based on the geographic distribution of the samples, nor with morphological characteristics previously described for the specimens of each species [3,94]. By contrast, the MSC species tree reconstruction (Figure 4, Appendix A) recovered three reciprocally monophyletic clades within the *S. flavius* + *S. libidinosus* group, with the first clade comprising samples of *S. libidinosus* from the Caatinga biome, the second comprising samples of *S. libidinosus* from the Cerrado, and the third comprising all putative *S. flavius* samples. Interestingly, all samples from the eastern Caatinga that were previously identified (based on morphological characters) as coming from the blond capuchin monkey were indeed recovered within the *S. flavius* clade in both the ML and MSC analyses.

### 3.3. Species Delimitation and Divergence Dating

We tested eight species delimitation model hypotheses using the BFD method (Figure 3). The rankings of the alternative models based on their Marginal Likelihood Estimates (MLE) and Bayes Factor Delimitation (BFD) are shown in Table 2. Model H7, which agrees with our MSC phylogenetic reconstruction, received “decisive” support over all other model hypotheses (2lnBF = 192.38–6867.94). Among the remaining models, model H5 was the next closest in rank, based on both MLE and BFD, while models H0 and H1 were the least favored species delimitation model hypotheses. Model H7 posits the existence of nine distinct lineages within the *Sapajus* radiation; these include *S. xanthosternos*, *S. robustus*, *S. nigritus*, *S. flavius*, and *S. cay*, plus two lineages within what has, heretofore, been called *S. libidinosus* (one lineage comprising samples from the Cerrado and the other comprising samples from the Caatinga) and two lineages within what has been called *S. apella* from the Amazon. Note, however, that in the Bayesian analysis, the three Amazonian lineages (*S. cay* and two lineages with *S. apella*) received only moderate bootstrap support (Figure 5), and one of the *S. cay* samples was recovered within one of the Amazonian clades.

Table 3 summarizes the results of the three different model calibrations used as priors in our StarBeast2 Bayesian divergence time analysis. The different root calibrations yielded variation in the posterior distribution of divergence times, with Model calibration 3 yielding the oldest estimates for all nodes, e.g., the split between the *Cebus* and *Sapajus* lineages [Node 2] as Median = 5.29 Ma; 95% Highest Posterior Density (HPD) interval= 1.95–7.58. In Figure 5, we present the divergence time estimates from this Model calibration 3 analysis.

## 4. Discussion

In this study, we used a phylogenomic approach to investigate the evolutionary relationships within the robust capuchin monkey radiation. Only a handful of prior investigations have used genetic evidence to evaluate phylogenetic relationships specifically within the genus *Sapajus* [5,13,95], and while these studies have provided some insight into the existing species diversity within the robust capuchin radiation, thus far, the picture has remained incomplete. Here, we used a larger sample size than that employed in any prior genetic study of the radiation, and we used a large set of phylogenomic markers to reconstruct the *Sapajus* phylogeny using both Maximum Likelihood (ML) and Multispecies Coalescent (MSC) methods. Importantly, we also applied a species delimitation approach within our MSC analysis, taking advantage of recent progress in genome-wide marker discovery and next generation sequence techniques, such as ddRADseq. Therefore, the results from this study provide new information to better understand species diversity within the robust capuchin monkeys and the evolutionary history among *Sapajus* lineages.

Both of our ML and MSC phylogenetic reconstructions provided support for the species status of *S. robustus*, the crested capuchin; *S. xanthosternos*, the yellow-breasted capuchin; and *S. nigritus*, the black-horned capuchin, corroborating previous findings [5,96] that also recovered these lineages as different taxa, though neither previous studies nor this study were able to confidently resolve the precise relationships among these species. Our findings support the placement of these three species from the Atlantic Forest of Brazil as the first splits within the robust capuchin radiation, with *S. nigritus* recovered as sister to all other *Sapajus* lineages from the Pantanal, Amazon, Cerrado, Caatinga, and northeastern Atlantic Forest regions. This finding disagrees with previous morphology-based taxonomies that have placed *S. robustus* as a subspecies of *S. nigritus* [11,12], but it corroborates other [3,7] taxonomic accounts for capuchin monkeys, including an earlier phylogeographic study that suggested an Atlantic Forest origin for the robust capuchins [97].

All of our analyses suggest that capuchin monkeys from the Pantanal and Amazon are divided into three reciprocally monophyletic clades. However, interestingly, one of the samples identified as *Sapajus cay* clustered with the *S. apella* 1 genetic cluster and the division of the samples within the two Amazonian clades do not agree with morphotypes or with the geographic division previously described for *S. apella* and *S. macrocephalus* species [3,7], as shown in Figure 6. Lima et al. (2018) [13] found no support for the molecular distinctiveness of the Pantanal and Amazon robust capuchin forms when using SNPs derived from Ultraconserved Element genomic markers (UCEs). These authors found some weak evidence for a northeastern and a southeastern clade within the Amazonian forms but also with the genetic lineages identified not agreeing with the current species hypotheses for the Amazonian *Sapajus*.

Silva-Júnior (2001) [3] analyzed morphological characters of more than 200 individuals of several localities throughout the Amazon for *S. apella* and 40 individuals from 12 localities for *S. macrocephalus* and described the distribution of both taxa based on these morphological characters while also considering the major rivers as possible barriers or limiting boundaries for these 2 Amazonian robust capuchin species. However, despite some evidence of rivers playing a role as a barrier for primate dispersal [98,99], other studies have demonstrated that, for some primate species, rivers might not, in fact, hinder animal movement and thus gene flow [100,101]. Therefore, even though our results do not support the geographic distributions previously inferred for two potential lineages of capuchins in the Amazon based on morphological analyses, they nonetheless demonstrate that two clades are indeed present: a “northeastern” clade potentially from the north of the Japurá and the Negro rivers at its most western portion in Brazil, widely overlapping with the previously described range for *S. apella*, and another “southwestern” clade to the south of these same rivers but extending through the Rondônia state of Brazil and somewhat overlapping with the previously described *S. macrocephalus* species range (Figure 6). Giving that these two lineages were recovered with strong support in both our ML and MSC analyses, and also considering the fact that these lineages were supported by the species delimitation approach used, we suggest that new studies are needed to better diagnose the potential morphological characteristics distinguishing these two lineages, the limits of their distributions (not only in Brazil but also in Colombia where potentially both clades might occur), and whether these lineages should be recognized as distinct subspecies.

In our study, too, Azara’s capuchin monkey, *S. cay*, was recovered as a sister clade to the rest of the widespread Amazon species cluster. However, only a small sample size for *S. cay* was used in this study (N = 3 individuals), and one of these samples clustered within the *S. apella* 1 clade, thus it is not possible yet to assess the taxonomic validity of *S. cay*. Still, it is interesting to note that both the northern range limits of *S. cay* and the southern limits of *S. apella* species are not completely defined, raising questions on the correct taxonomic identification of the one *S. cay* sample that clustered within the Amazonian clade.

Overall, our phylogenetic reconstructions suggest that putative *Sapajus libidinosus*, sampled from across the dry biomes of the Caatinga and Cerrado, are paraphyletic and fall within a widespread clade with a geographic range that spans the Cerrado, Caatinga, and northeastern Atlantic Forest and includes all samples putatively from both *S. flavius* and *Sapajus libidinosus*. MSC phylogenetic reconstruction recovered two reciprocally monophyletic clades within putative *S. libidinosus*, corresponding to the geographic distribution of the samples, with one genetic cluster composed of the samples from the Cerrado, while the second corresponded to the samples from the Caatinga. In our ML analysis, the putative blond capuchin monkey samples were recovered as clustered within the set of samples of *S. libidinosus* from the eastern Caatinga. Contrarily, both of our MSC phylogenetic reconstructions recovered *S. flavius* as a monophyletic clade, with the SVDquartets analysis recovering the blond capuchin as sister to the *S. libidinosus* clade from the Cerrado, and our Bayesian phylogenetic reconstruction instead suggesting that *S. flavius* is sister to the *S. libidinosus* clade from the Caatinga, both with strong support. Species delimitation analysis gave decisive support to model hypothesis H7, which considered three species for the capuchin monkeys within the Cerrado, Caatinga, and northeastern Atlantic Forest, corroborating the MSC phylogenetic reconstruction.

The fact that both our MSC and species delimitation analyses found, with strong support, that the widespread robust capuchins occupying the Cerrado and Caatinga belong to two reciprocally monophyletic clades challenges the current taxonomy for both *S. libidinosus* and *S. flavius*. Phylogenetic reconstruction incongruences, such as those that we found among the capuchin monkeys from the Cerrado, Caatinga, and northern Atlantic Forest, could indicate either the finding of a new cryptic lineage in the Caatinga biome or, rather, true inconsistencies due to incomplete lineage sorting or hybridization, for example. Incomplete lineage sorting (ILS), a process by which ancestral polymorphisms can persist through species divergences, and gene flow across species boundaries caused by introgressive hybridization, might generate gene tree discordances, hampering species tree estimation [102,103,104]. While MSC phylogenetic approaches have improved model complexity, making it possible to specifically account for lineage sorting and intraspecific variation within individuals [105], such models cannot account for high levels of gene flow, which has been shown to affect species tree inferences by decreasing posterior clade probabilities, underestimating divergence time estimates, and altering the species tree topology [102].

Phylogenetic relationships among taxa within the *Sapajus* radiation have continued to be contentious, with some studies suggesting *Sapajus* is a recent evolving lineage characterized by a high degree of past gene flow among certain lineages as well as ongoing admixture [5,13,106]. In addition, more recent phylogenetic and phylogenomic studies have supported recognizing either four or six species within the *Sapajus* radiation [5,13]. In this study, we applied a species delimitation method to explicitly evaluate different model hypotheses for the number of species in the robust capuchin genus and to identify potential evolutionarily independent lineages (e.g., distinct species). Our Bayes Factor species delimitation analysis suggests that the *Sapajus* radiation is composed of nine distinct lineages: *S. xanthosternos*, *S. robustus*, *S. nigritus*, *S. libidinosus* cluster 2 (samples from the Cerrado biome), *S. libidinosus* cluster 1 (samples from the Caatinga), *S. flavius*, *S. cay*, *S. apella* cluster 1 (corresponding to *S. macrocephalus*), and *S. apella* cluster 2. Overall, this result agrees with the morphology-based taxonomies of Silva-Júnior (2001) [3] and Rylands and colleagues (2013) [7], except that it highlights the paraphyly of *S. libidinosus* and suggests the geographic distributions of both Amazonian capuchin monkeys need to be reconsidered.

Importantly, this study has filled a longstanding gap regarding sample collection from both the eastern *S. libidinosus* and westernmost *S. flavius* ranges, corroborating previous morphology-based studies that suggested the presence of blond capuchin monkeys in some areas of the dry Caatinga [16,94], as all the samples putatively identified, based on morphology, as *S. flavius* from the Caatinga was indeed clustered with blond capuchin samples from the Atlantic Forest in all our analyses. However, in light of this new genetic evidence for the presence of the blond capuchin monkey in the Caatinga, as well as genetic evidence of hybridization among *Sapajus* lineages [13], additional analyses are needed to further investigate whether the paraphyletic arrangement found in the *S. libidinosus* samples represents a new genetic lineage within the Caatinga or, rather, reflects gene flow and introgression between the bearded capuchin monkey and the blond capuchin in the transition areas between the Atlantic Forest and Caatinga biomes.

Identifying species limits has never been a straightforward task [107]. Therefore, to robustly infer species boundaries, the use of integrative taxonomy has become increasingly common. Although sampling markers from across the genome and using models that account for different coalescent histories and discordance among loci can provide more objective measures for assessing species relationships and delimiting taxa [19,69,70,78,79], species delimitation approaches, such as the one used in this study, should be seen as one source of evidence, which should be analyzed along with other lines of evidence as much as possible [107,108]. Therefore, new studies are necessary to better understand capuchin monkey species distributions and the occurrence of hybridization. Additionally, it will be important to further increase sampling of species and areas still poorly represented in this and other previous studies.

Finally, the time tree generated from our Starbeast2 analysis (Figure 5, Table 3), calibrated with the upper bound time estimate for the divergence of *Saimiri* and the capuchin monkeys, *sensu lato* (Model calibration 3), placed the estimated divergence time for *Cebus* and *Sapajus* genera at 5.29 Ma. While this is the oldest of the divergence times estimated for this particular node in our study, it is nonetheless more recent than the mean divergence time estimated between gracile and robust capuchin monkeys in other recent studies (5.8 Ma [5]; 6.6 Ma [13]; 6 Ma [109]; and 6.6 Ma [110]), although it is close to the divergence time estimated by [101], at 5.39 Ma based on whole mtDNA genomes. This divergence time estimate corresponds to a late Miocene divergence time for *Cebus* and *Sapajus* genera, which agrees with the hypothesis of the savanna-like environments in the Cerrado favoring a vicariance event separating primate populations of different genera, including populations of a capuchin ancestor occurring throughout the Amazon and Atlantic Forest [111]. The divergence time estimates for all other nodes within the *Sapajus* genus were more recent than those estimates found by [13], indicating the reinvasion of *Sapajus* into the Pantanal, Amazon, Cerrado, Caatinga, and northeastern Atlantic Forest was a recent event, occurring only 0.6 to 1.7 Ma, based on the divergence time estimated for the split between the *S. nigritus* and all other robust capuchin lineages outside the Atlantic Forest to the south of the São Francisco River.

## 5. Conclusions

In this study, we used the largest sample size to date to study the evolutionary history of the *Sapajus* genus and successfully generated data matrices with thousands of genomic markers for all putative species of robust capuchin monkeys. All of our analyses (ML and MSC phylogenetic reconstructions as well as species delimitation model testing under Bayesian inference) were congruent regarding the evolutionary history of the species from the Atlantic Forest south of the São Francisco River and of the species occurring in the Pantanal and Amazon in Brazil. *Sapajus robustus*, *S. xanthosternos*, and *Sapajus nigritus* were recovered as three monophyletic clades and as the first splits in the robust capuchin radiation, with *S. nigritus* recovered as closer to all other robust capuchin lineages. In addition, the Pantanal and Amazonian *Sapajus* were recovered as being structured into three monophyletic clades, although *Sapajus cay*, Azara’s capuchin monkey, only received strong support from one (ML) out of three phylogenetic reconstructions, while the division of the Amazonian capuchin monkeys into two reciprocally monophyletic clades was strongly supported by two (ML and MSC) out of three phylogenetic reconstructions. We suggest these two clades should be considered at least as valid subspecies, with *Sapajus apella apella* as the lineage occurring to the north of the Jupará and Negro rivers and extending as a northeastern clade and with *Sapajus apella macrocephalus* as the lineage occurring in the southwestern Amazon to the south of the Negro River. However, new morphological assessments are necessary, as these Amazonian clades do not agree with previous morphology-based taxonomic distributions for the Amazonian capuchin monkeys.

Our phylogenetic reconstructions for the capuchin monkeys occurring in the Cerrado, Caatinga, and northeastern Atlantic Forest were less congruent, as we recovered the bearded capuchin monkey as a paraphyletic clade, with samples from the Caatinga biome belonging to either a monophyletic clade (MSC) or grouped with samples of the blond capuchin monkey (ML). Despite the strong support from both the MSC and the species delimitation approaches, further analyses are necessary to indicate whether this incongruence regarding the placement of the *Sapajus libidinosus* samples from the Caatinga is due to the occurrence of gene flow. Finally, our species delimitation approach supported the division of the robust capuchin monkey into nine different species. However, this result should be seen as a taxonomic hypothesis and, as such, subject to further testing.

## Figures and Tables

**Figure 1 genes-14-00970-f001:**
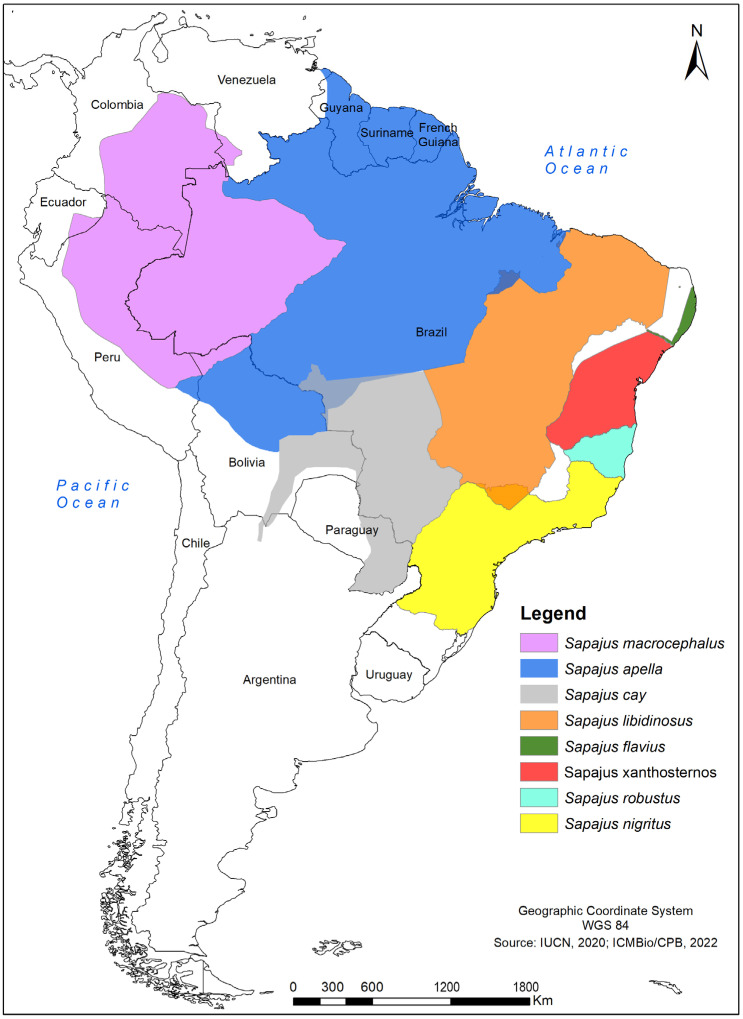
Map with the potential distributions of the *Sapajus* species according taxonomic arrangement with eight species [3,15].

**Figure 2 genes-14-00970-f002:**
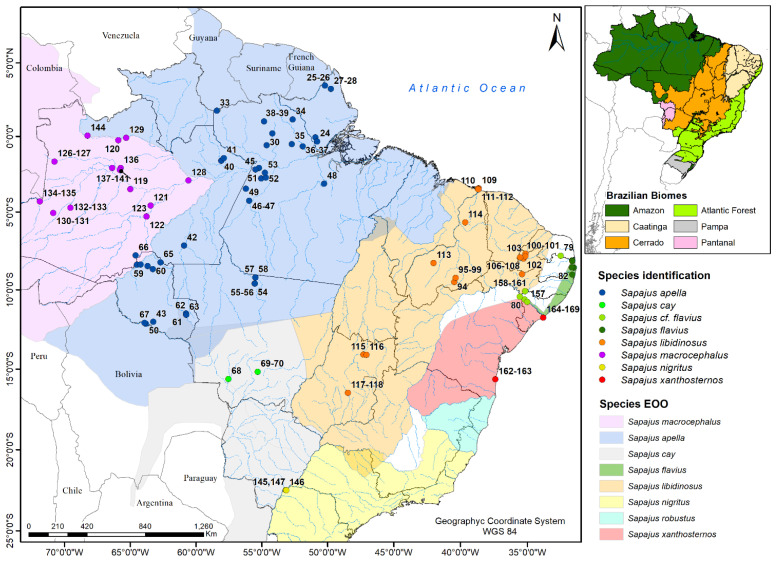
Map showing the sampled localities. Sample numbers correspond to those in Appendix A. *Sapajus* cf. *flavius*—samples from populations of capuchins occupying areas of the Atlantic Forest–Caatinga transition whose taxonomic assignment as *S. flavius* is uncertain.

**Figure 3 genes-14-00970-f003:**
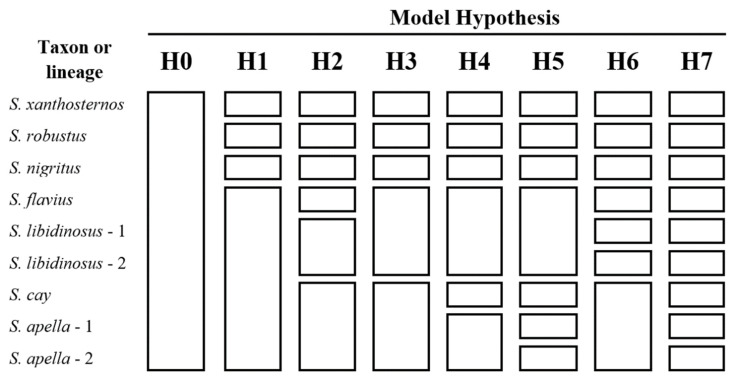
Species delimitation model hypotheses (H0 to H7). Each species delimitation model has a specific combination of lineages (rows).

**Figure 4 genes-14-00970-f004:**
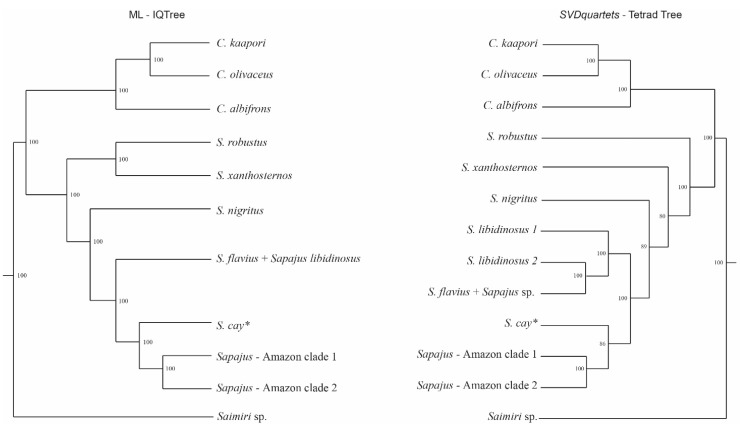
Maximum likelihood (ML-IQ-TREE, **left**) and SVDquartets species tree (SVDquartets—Tetrad tree, **right**) phylogenies for *Sapajus* radiation. Numbers at each node represent bootstrap support values (based on 1000 replicates). * one of the samples putatively assigned to *S. cay* was recovered within one of the Amazonian clusters. See Appendix A for the complete trees.

**Figure 5 genes-14-00970-f005:**
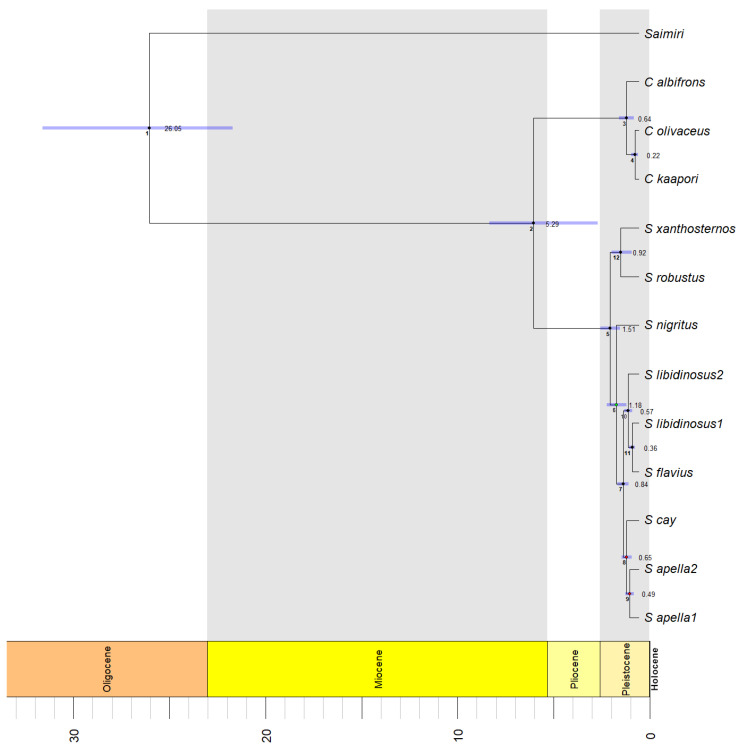
StarBeast2 Bayesian divergence time analysis with node heights scaled to median divergence time estimates. Small numbers closer to the nodes correspond to the node labels in Table 3; numbers at the right of each node indicate the posterior means of the ages for each node. Blue bars represent 95% Highest Posterior Density (HPD) intervals. Colors of the nodes indicate the support values: black = high support/posterior ≥ 0.99; green = moderate support/posterior ≥ 0.95 < 0.99; red = low support/posterior < 0.95 (see Appendix A for detailed support values for each node).

**Figure 6 genes-14-00970-f006:**
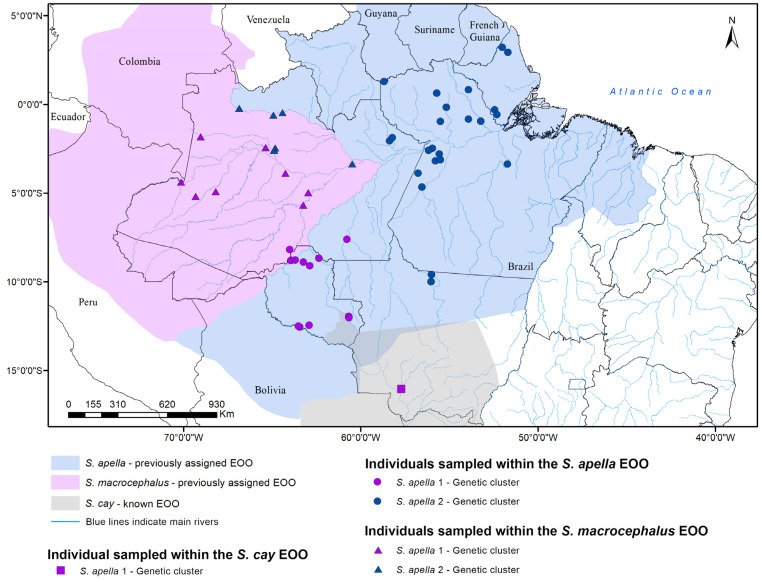
Map showing the genetic clusters found for Amazonian robust capuchin monkeys in both ML and MSC phylogenetic reconstructions, the previously assigned Extension of Occurrence (EOO) for *S. macrocephalus*, *S. apella*, and *S. cay,* and morphology-based identification of individuals samples, according to [3].

**Table 1 genes-14-00970-t001:** Total number of loci and the total number of SNPs included in the final genotype matrices, based on the minimum number of samples per locus for the output matrix.

Minimum % to Call a Locus ^1^	Minimum # to Call a Locus ^2^	Number of Loci ^3^	SNPs Matrix Size	% Missing SNPs	Total # of Variable Sites	Parsimony- Informative Sites ^4^	% Missing Sequence Matrix
≈90%	150	16,880	353,995	15.38%	327,622	178,012	20.4%
≈75%	130	31,216	614,050	19.34%	572,259	299,054	24.2%
≈60%	100	43,736	830,442	24.41%	777,155	394,844	29.2%
≈30%	50	64,081	1,125,828	34.32%	1,054,796	518,460	40.1%

^1^ Minimum number of samples a locus must be present in to be included in the final genotype matrix, represented as a percentage of the total number of samples. ^2^ Number of samples used as parameter for ipyrad. ^3^ Number of loci in the final genotype matrix. ^4^ Total number of parsimony-informative sites estimated by IQ-TREE.

**Table 2 genes-14-00970-t002:** Summary of results from the Bayes Factor species delimitation analysis. Species delimitation models (H0–H7) ordered by rank, their marginal likelihood estimates (MLE), and Bayes Factor testing results (2lnBf) from the analyses with the Path Sampling (PS) methods.

Model Hypothesis ^1^ (Ranked by MLE)	Marginal Likelihood Estimate (MLE)	Bayes Factor (2lnBF) ^2^ to the 1st Ranked Model H7	Bayes Factor (2lnBF) ^2^ to the 2nd Ranked Model H5
H7	−185,731.7644	-	−192.38
H5	−185,827.9526	192.38	-
H6	−186,006.9226	550.32	357.94
H1	−186,153.4149	843.30	650.92
H4	−186,337.5626	1211.60	1019.22
H3	−186,424.9737	1386.42	1194.04
H1	−187,530.4374	3597.35	3404.97
H0	−189,165.7362	6867.94	6675.57

^1^ Model hypothesis as shown in Figure 3. ^2^ Bayes Factor, calculated by multiplying twice the ratio of the MLE of one model by the MLE of a competing model [BF = 2 × (MLEHx − MLEHy)], according to [79].

**Table 3 genes-14-00970-t003:** Summary of the posterior distribution of divergence times (in Ma) estimated using the software Starbeast2.

Node in Time Tree ^1^	Model Calibration 1		Model Calibration 2		Model Calibration 3 ^2^	
	Median	95% HPD	Median	95% HPD	Median	95% HPD
1	13.04	12.1–16.39	14.03	13.08–15.46	26.05	21.71–31.62
2	2.85	0.9–3.89	3.08	1.26–3.83	5.29	1.95–7.58
3	0.34	0.13–0.53	0.37	0.16–0.53	0.64	0.26–1.03
4	0.11	0.02–0.2	0.12	0.02–0.21	0.22	0.04–0.41
5	0.78	0.52–1.07	0.83	0.57–1.04	1.51	0.99–2.02
6	0.61	0.34–0.88	0.65	0.4–0.88	1.18	0.66–1.69
7	0.43	0.29–0.59	0.46	0.33–0.58	0.84	0.56–1.14
8	0.33	0.21–0.47	0.35	0.23–0.48	0.65	0.39–0.9
9	0.25	0.13–0.37	0.26	0.16–0.38	0.49	0.28–0.72
10	0.29	0.18–0.4	0.32	0.21–0.39	0.57	0.35–0.75
11	0.19	0.1–0.27	0.2	0.13–0.26	0.36	0.21–0.51
12	0.5	0.17–0.74	0.53	0.2–0.72	0.92	0.36–1.42

^1^ See Figure 5 for node labels. ^2^ Posterior distribution of divergence times shown in Figure 5.

## Data Availability

Not applicable.

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
