# Peer review of "A New Assessment of Robust Capuchin Monkey (*Sapajus*) Evolutionary History Using Genome-Wide SNP Marker Data and a Bayesian Approach to Species Delimitation"

_genes, 2023, doi:10.3390/genes14050970_

Round 1

Reviewer 1 Report

This study describes the use of SNP marker data to cope with the taxonomy, systematics, and evolutionary history of robust capuchin monkey (genus Sapajus) which, given increasing inconclusive evidence, has yield a shifting and rather unstable taxonomy. To achieve this, the authors take advantage of a number of state-of-the-art phylogenetic methods, such as maximum likelihood, multispecies coalescent phylogenetic inference, and the Bayes Factor method to test for alternative hypotheses of species delimitation. The paper concludes that Sapajus radiation led to three species from the Atlantic Forest south of the São Francisco River, and the Pantanal and Amazonian Sapajus are structured into three monophyletic clades. However, the taxonomic distributions of the Amazonian clades do not agree with previous morphology-based classifications, suggesting that further morphological assessments are necessary. The phylogenetic reconstructions for Sapajus occurring in the Cerrado, Caatinga, and northeastern Atlantic Forest were less congruent, with the bearded capuchin being recovered as a paraphyletic clade, and samples from the Caatinga biome being either a monophyletic clade or nested with the blond capuchin monkey. Overall, the paper seems to provide a detailed analysis of the evolutionary history and taxonomy of the Sapajus genus, with the use of genomic data providing important insights into the relationships between different species and populations.

The manuscript presents potentially interesting information, and in my opinion, it only needs minor revision. Here, I provide very few comments and observations.

Typos and minor:

Lines 102-107- It seems that the authors will “shed light on the mechanisms influencing a rapid diversification of lineages”, but this is not actually addressed in the manuscript. I think this sentence should be toned down.

Reviewer 2 Report

This manuscript implemented ddRADseq approach to generate genome-wide SNP markers for individuals from all putative species of Sapajus to infer their phylogenetic history, and evaluate the number of discrete species. Their results suggest that the Sapajus radiation is composed of nine distinct lineages: S. xanthosternos, S. robustus, S. nigritus, S. libidinosus cluster-2 (samples from the Cerrado biome), S. libidinosus cluster-1 (samples from the Caatinga), S. flavius, S. cay, S. paella cluster-1, and S. apella cluster-2 (corresponding to S. macrocephalus).

The taxonomists and phylogeneticists have long been controversial in capuchin taxonomy, the authors resolve this issue by identifying nine species. There are several concerns that should be addressed:

1) The phylogenetic relationships between S. xanthosternos and S. robustus, and between S. libidinosus and S. flavius are conflicting in ML and MSC analyses. The results are not robust in MSC analyses while the support values for the split of the monophyletic clade comprising either S. xanthosternos or S. nigritus were moderate, In the Bayesian analysis, the three Amazonian lineages (S. cay, and two lineages with S. apella) also received only moderate bootstrap support. The phylogenetic relationships that are not well-supported make it questionable to design species identification models based on the trees.

2) TABLE 1. The sample information, such as Saimiri, Sapajus cf. flavius, is confusing. I would suggest to add the information that the broad “eight species” classification from previous studies [7,10]. This is also the case for Figure 1 and Figure 2.

3) Line 199: “For most newly collected samples” changed to Samples newly collected in this study.

4) Line 206: “Most of the samples yielded sufficient genomic” changed to The newly yielded genomic”.

5) Lines 277-279: Four progressively smaller data matrices instead of three.

6) Figure 3. For the species delimitation model hypothesis, why not test the model of Sapajus libidinosus-2 and S. flavius as one species based on the results of MSC analysis.

7) Lines 419-420: What about the results of the other data sets? How about the support values?

8) Lines 423-445: The two clades, northeastern and southeastern clade, within Amazonian need to be described in the Introduction.

9) Line 482: “moderate bootstrap support” need show detail support value in the tree of Figure 5.

10) Lines575-577: The same figure legend with Lines 571-574.

11) The result of divergence times based on different model calibration in Table 4 also need to be discussed. Why the result of model calibration 3 is used?

Round 2

Reviewer 2 Report

The authors have addressed my concerns

Author Response

Dear Reviewer,

Thank you for commenting on our paper, “A new assessment of robust capuchin monkey (Sapajus) evolutionary history using genome-wide SNP marker data and a Bayesian approach to species delimitation.”